# The War after War: Volumetric Muscle Loss Incidence, Implication, Current Therapies and Emerging Reconstructive Strategies, a Comprehensive Review

**DOI:** 10.3390/biomedicines9050564

**Published:** 2021-05-18

**Authors:** Stefano Testa, Ersilia Fornetti, Claudia Fuoco, Carles Sanchez-Riera, Francesco Rizzo, Mario Ciccotti, Stefano Cannata, Tommaso Sciarra, Cesare Gargioli

**Affiliations:** 1Department of Biology, Rome University Tor Vergata, 00133 Rome, Italy; stefanotesta87@hotmail.it (S.T.); ersiforn@hotmail.it (E.F.); claudia.fuoco@uniroma2.it (C.F.); carles.sanchezr@gmail.com (C.S.-R.); cannata@uniroma2.it (S.C.); 2Joint Veteran Center, Scientific Department, Army Medical Center, 00184 Rome, Italy; fracesco.rizzo@tim.it (F.R.); mario.ciccotti@yahoo.it (M.C.)

**Keywords:** VML, war muscle injuries, permanent disability, prosthesis, reconstructive therapies, muscle loss, war, muscle injury

## Abstract

Volumetric muscle loss (VML) is the massive wasting of skeletal muscle tissue due to traumatic events or surgical ablation. This pathological condition exceeds the physiological healing process carried out by the muscle itself, which owns remarkable capacity to restore damages but only when limited in dimensions. Upon VML occurring, the affected area is severely compromised, heavily influencing the affected a person’s quality of life. Overall, this condition is often associated with chronic disability, which makes the return to duty of highly specialized professional figures (e.g., military personnel or athletes) almost impossible. The actual treatment for VML is based on surgical conservative treatment followed by physical exercise; nevertheless, the results, in terms of either lost mass and/or functionality recovery, are still poor. On the other hand, the efforts of the scientific community are focusing on reconstructive therapy aiming at muscular tissue void volume replenishment by exploiting biomimetic matrix or artificial tissue implantation. Reconstructing strategies represent a valid option to build new muscular tissue not only to recover damaged muscles, but also to better socket prosthesis in terms of anchorage surfaces and reinnervation substrates for reconstructed mass.

## 1. War-Related Skeletal Muscle Injuries

The musculoskeletal system occupies the largest volume of the body, revealing the ability to self-regenerate upon tissue damage. However, its regeneration ability is limited in relation to the extent of the damage, i.e., muscular tissue fails to regenerate injuries such as extensive loss of mass, also known as volumetric muscle loss (VML) [1].

VML includes both traumatic lesions and surgical removals of a large muscle area with the associated loss of stem cells and extracellular matrices (ECMs) and the consequent impairment of regenerative capabilities [2]. VML is a common, pathological condition that may occur in relation to primary trauma such as crush injuries, penetrating trauma and blasts, or secondary trauma-like compartment syndromes and comorbidity to open bone fracture [3,4]. However, the frequency of VML injuries in civil and military population is disproportionate, with the latter being the most affected group. In fact, it has been estimated that among 14,500 military personnel evacuated from battlefields from 2001 to 2013, 77% reported musculoskeletal injuries [5].

In modern conflicts, injuries to limbs cover the majority of battlefield wounds. Corona and colleagues [6] collected data during the first years of the 2000s from both Operation Enduring Freedom (OEF) and Operation Iraqi Freedom (OIF). In particular, the authors analyzed a group of retired service members with orthopedic trauma (type III tibia fracture) and a group of battlefield-injured service members who retired due to various injuries (general). In both cases, among service members medically discharged from their military occupations, muscle-related disabling conditions were reported. In particular, in the orthopedic group, 65% were affected by VML, a value higher than the general group reaching 92% of muscle-related injuries [6].

Akpoto and collaborators [7], in a study conducted from 1st May to 31st December 2014 (8 months) during the Mali conflict, examined war-related extremity injuries in 50 soldiers recovered at Togo Level two Hospital. Interestingly, most of war-related traumas were due to sudden explosive devices (36%), followed by mortar and rocket shrapnel (30%). The lesions were distributed mainly on the lower extremities (62.92%), and soft tissue wounds (skeletal muscle was the most representative) were the second most common accidents during the conflict (28.09%).

Another important complication of war trauma is heterotopic ossification (HO). This process is mediated by the complex interaction of both systemic and local wound inflammation that, upon an intricate cascade of events, leads to the formation of lamellar bone in non-osseous tissues [8,9,10]. In particular, the wound stimulate mesenchymal stem cells (MSCs) mobilization in a injury site; here the local released growth factors (BMP family) and inflammatory cytokines (IL-3, IL-6, and IL-10) promote the differentiation of MSCs in osteoprogenitors through the Runx-2-mediated pathway [11]. HO can occur in both civilian and military trauma, but the incidence is unpaired: for civilian trauma, HO formation is due to the combination of a central nervous system (CNS) trauma and mainly a fracture of the femur (54% of patients with thigh trauma) [12]. In war trauma, HO occurs with or without CNS trauma, with a high incidence rate that reaches 64%, following military blast injuries [13,14]. Nowadays, the only effective therapy for HO is surgical removal that has to be ideally performed at least six months after the injury occurs to allow for an adequately ectopic bone cortication and maturation [15]. Unfortunately, this approach generates VML in skeletal muscle tissue, inevitably compromising the functionality of the interested area. For these reasons, HO pathology represents the most significant obstacle to independence, functional mobility, and return to duty for combat-injured veterans [16].

## 2. Social Implication of Muscle-Related Disabilities

Extremity injuries are one of the most common wounds among military personnel in combat operations and involve bone, muscle, and surrounding tissues. These traumatic events often lead doctors to the sad reality of having to choose between salvaging a limb versus amputation. The number of people amputated worldwide is relevant, although it is difficult to obtain a reliable estimate regarding limb amputation given that not all countries regularly register the cases: it is estimated that 150,000 people are admitted to hospitals each year to undergo amputation with many cases resulting from complications in diabetes [17]. In 2005, estimated 1.6 million people in the U.S. were amputated, with 65% of them having lower limb amputations [18]. Worldwide, estimated 40 million amputees are present, resulting in 30% with arm amputations, of which 2.4 million in developing countries have the rate of 59% carrying below elbow amputations, 28% carrying above elbow and elbow disarticulation, 8% carrying shoulder, and 5% carrying hand/wrist amputations [19]. In a recent study on physical and psychological health in military amputees during rehabilitation, Talbot and colleagues noticed that, whereas physical health improved as might be expected upon rehabilitation, mental health component did not [20]. As the physical and mental components are interacting constantly during the rehabilitation process, particular attention is needed about mental health. However, improvements in modern surgical procedures allows the limb to be saved. Nevertheless, for patients with extensive injuries, issues do not end at the acute treatment phase, and most of them require long-term rehabilitation and post-acute services to optimize recovery, allowing reintegration into military service and daily life. Moreover, during the years following injury, the risks to develop secondary adverse health effects (e.g., depression, obesity, and chronic pain) are high and could heavily affect the patient’s quality of life, especially younger ones [21,22,23].

When reintegration is not possible, the soldier is medically retired, and the conditions (physical/psychical) that determine the inability to return to active duty are considered as disqualifying service and disabling conditions. Among these, VML contributes the most to long-term disability of military personnel [6]. In fact, surprisingly, preclinical models of VML show that a relatively small VML (loss of muscle: <20%) can lead to a significant strength deficit (30%–90%) [3,24,25,26]. For this reason, a moderate muscle loss could lead to a disabling condition.

Beyond the catastrophic impact of this kind of muscle-related disability on patients’ daily lives, the loss of a highly specialized professional figure is a double-hit for the army, i.e., the loss of investment for the formation and the specialization of the soldier and high costs of medical treatment. As observed by the above cited study [6], based on 2014 U.S. Military Pay Charts, the costs per individual ranged from $340,000 to $440,000. To note, these values do not take into account medical costs, loss of lifetime wages, or Veteran’s affairs disability-related costs [4].

In civil trauma, VML injuries are not well tracked, although it is estimated that in the U.S. alone, there are approximately 150,000 open fractures occurring each year and among these, the majority are associated with the loss of surrounding soft tissue [27], without taking into account sport injuries and muscle tissue loss upon cancer surgical removal [28]. Moreover, about 50% of patients with severe lower extremity injuries are unable to return to work because of the development of chronic disability [29]. This situation is common for civilian and military patients, demonstrating the heavy impact of VML injuries on the society.

## 3. VML Diagnosis Techniques and Prosthesis-Related Problems

Although great progresses in military medical science have been made regarding VML in the last 20 years, there are still some issues that complicate the treatment of these diseases. One of the greatest problems of VML in military field is to face up an appropriate diagnosis. A well-done diagnosis includes an accurate quantification of the damage allowing a double benefit: (i) the determination of an appropriate intervention by the medical staff; (ii) a precise follow-up upon the recovery. In recent years, four main techniques have been commonly used to estimate damaged muscle mass: bioelectric impedance (BIA), dual energy X-ray absorptiometry (DXA), computed tomography (CT), and magnetic resonance imaging (MRI) [30,31,32].

After an accurate study of these methods and possibilities of implementing them in battlefield hospitals, Buckinx and colleagues made the conclusion that DXA, based on the feasibility (accuracy, safety, and low cost), could be considered as the referenced standard for evaluating muscle mass [33].

Another issue concerning VML regards prosthetic applications that are closely related to the extension of muscle trauma and the quality of residual tissue. However, initial solutions are more commonly found for VML in the lower extremities, which is still a field that needs further investigation given the large number of people involved [19]. Concerning VML in the upper extremities, there are many aspects that need to be explored in depth, starting a novel research line regarding the choice of prosthesis and reinnervation techniques in relation to the dimension of the muscle contact surface and the prosthesis itself. Moreover, myoelectrically controlled prostheses need innerved skeletal muscle tissue for functioning and then for enabling the movements of artificial limbs. In fact, throughout the targeted muscle reinnervation technique, nerves of brachial plexus (upper limb) are moved from their original locations to stimulate specific areas of residual muscles in stumps. These areas are used as “switches” to produce surface electromyography (EMG) signals, which are recorded by the prosthesis’ sensors, allowing the execution of precise movements like bending the forearm, turning the palm, and/or closing the hand [34]. Nevertheless, for this kind of approach, an essential condition is the presence of functional skeletal muscle tissue in the prosthesis socket, a condition that is difficult to satisfy when VML occurs. However, this approach presents intrinsic limitations that have to be overcome; in fact, to provide a natural control of prosthesis with multiple degree-of-freedoms (DOFs) based on multi-channel EMG signals, subjects need intense training to consciously manage fine movements of a prosthesis. This continuous effort brings most of them to feel fatigue after a prolonged use of an artificial limb [35]. An alternative source of signals to control a prosthesis could be taken by ultrasound technology. In fact, the use of ultrasounds to both stimulate muscle mass [36] and record fine muscle architecture changes is well documented in the literature [37,38,39], but not yet very utilized. In particular, the continuous real-time monitoring of muscle geometry changes by an ultrasound image is called sonomyography (SMG) and could be used for prosthetic control. Changes in muscle thickness captured by SMG are used to start specific movements of a prosthesis; and, the first studies have demonstrated a significantly higher accuracy with SMG compared to that with EMG control systems [40]. In the same way, for SMG-controlled prostheses, the essential requirement is the presence of functional skeletal muscle tissue on the stump, with a volume as comparable as possible to the native muscle, which is a condition following a traumatic event such as blasts or other traumatic injuries difficult to observe.

## 4. Current Treatments of VML

### 4.1. Surgical Treatments

VML injuries occur following a large variety of trauma ranging from high-energy injuries, such as blasts or bullet wounds, to crush injuries as a consequence of an open fracture (e.g., tibia). The latter happens in both daily life and battlefields, and in many cases, is accompanied by soft tissues loss. This type of trauma results in two different possible outcomes: limb savage and amputation. Moreover, even in cases where the limb is saved, the functional recovery is not entirely completed, leading to chronic disability in the worst cases. The clinical approach for open tibial fractures (one of the major causes of an orthopedic pathological condition associated with VML [6]) consists in a rapid wound debridement, followed by primary stabilization and injury cover. In a recently published work [41], a study on 33 patients affected by open tibial fractures is reported. In particular, the importance of acting quickly is highlighted, performing simultaneously during surgical operation the following steps: debridement, external fixation of bone elements, and soft tissues reconstruction using free distant/local rotational muscle flaps and/or fasciocutaneous flaps and wound coverage. Patients were surveyed and tracked for up to three years, and good outcomes were obtained.

However, replacing lost muscle tissue still remains unachievable, even using free flaps. In these cases, the functional recovery of VML is heavily impaired. In a study conducted by Garg and colleagues [42], the case of a 23-year-old soldier with a type III open tibia fracture due to a blast is reported. As a consequence of the trauma, the VML of both anterior and posterior compartments of the involved leg occurred. Once healed, after interventions for saving the limb and fracture stabilization (approximately 18 months post-injury), functional capacities were assessed. In particular, isometric and isokinetic analysis revealed a strong dorsi/plantar flexor deficit (34 over 100%). This is the case which demonstrates a reconstructive therapy could be crucial for the restoration of lost soft tissue (especially skeletal muscle) and its functionality.

### 4.2. Reconstructive Therapies

In clinical practice, despite limitations in the use of human stem cells, the use of an ECM as an acellular approach to VML treatment has been investigated over the last years. In fact, the biological function of an ECM in recreating a suitable microenvironment niche, which is able to promote the healing process toward a reconstructive and functional outcome, is intensively studied [43,44]. In a study performed by Sicari and colleagues [45], five patients with documented VML (three military persons and two civilians), previously subjected to multiple surgical procedures without benefits, were chosen. All of them exhibited a minimum of 25% functional and structural deficit. Accordingly, the patients were subjected to reconstructive interventions with a porcine urinary bladder-derived ECM. In particular, the procedure consisted in the removal of scar tissue, the identification of adjacent vascularized and innervated muscle tissue, and the implantation of a xenogenic ECM. The patients were followed for up to six months, and both immune-histological and functional tests were conducted. Interestingly, the immune-histological analysis on tissue biopsies revealed a consistent presence of perivascular stem cells (PVSCs) in neo-formed tissue, located not only in the proximity of blood vessels, thus indicating a possible active role of these types of cell population in muscle fibers formation. Moreover, the functional analysis showed partial strength recovery for three out of five patients. However, five patients are a statistically unsatisfactory sample size. In [46], the authors used the same approach for a group of 13 patients, with an average of 66% tissue deficit. In addition, in these cases, the reconstructive procedure consisted in biomimetic scaffold implantation (ECM from porcine urinary bladder), followed by an intensive and early physical therapy treatment. The relative immune histological analysis highlighted the central role of PVSCs in vascularization and innervation during skeletal muscle islands formation within the implantation site. The functional analysis showed an average improvement of 37% in strength and that of 27% in range of motion, representing still an unsatisfactory reconstructive outcome.

## 5. Preclinical Reconstructive Therapies for VML Treatment

Preclinical studies on tissue-engineering-based approaches to VML restoration have been conducted, over the last decades, on animal models. In particular, several cell populations were investigated to test their myogenic potential and their appeal in cell culture [47], as well as different biological scaffolds and their resulting combinations. In small animal models, like rodents, the obtained results are substantial [48], but the real challenge today is in restoring VML injuries on large animal models, like pigs or sheep, to lay the foundation for promising clinical trials.

### 5.1. Small Animal Models

Investigating the engineered, skeletal muscle constructs the capability to restore a massive area of removed skeletal muscle tissue on small animals, such as mice and rats; it is the first step in developing an appealing therapy for human injuries. For this reason, over the last years, many research teams have focused their efforts on this aim [49]. Wang and colleagues [50] used an approach based on collagen I scaffold and murine-muscle-derived stem cells to restore VML injuries done on rectus femoris. Implanted mice were followed and documented for eight weeks, with micro-computed tomography analysis that highlighted the muscle volume increment in the implanted mice compared with that in the control. The immune histological analysis after eight weeks confirmed the formation of organized muscle fibers within the defect area in the implanted mice only. Moreover, Fuoco and colleagues [48] studied the potential of a PEG-Fibrinogen (PF)-based biomimetic matrix combined with murine mesoangioblast as a cellular component for muscle reconstruction in a tibialis anterior (TA) VML mouse model [51,52,53]. In particular, immediately after surgical excision of about 90% of TA, the VML damage was replenished with the in situ casting of an artificial construct derived from PF and Mesoangioblast (Mabs). The mice were followed and documented for up to six months and exhibited a complete, functional recovery, supported by the immune histological analysis that showed the formation of mature skeletal muscle tissue properly vascularized and innervated. Interestingly, in both papers, the implanted stem cells survived in the regenerated muscle, but there was a strong presence of host cells, revealing a key role of the host during the regeneration process. In a study conducted by Quarta and collaborators [54], the importance of different cellular components in engineered bio-constructs was demonstrated. The authors isolated muscle satellite cells (MuSCs) and muscle resident cells (MRCs), with the latter being a heterogeneous population composed of hematopoietic cells, endothelial cells, fibro adipogenic progenitors (FAPs), and fibroblast-like cells, by choosing an ECM-based hydrogel as the scaffold. Subsequently, bio-constructs composed of either MuSCs or a combination of MuSCs and MRCs were implanted in a VML mouse model, designed by 40% of TA removal resulting in a 40%–50% force reduction. Interestingly, the best results in terms of force restoration, muscle fibers formation, and vascularization were obtained with the combination of the two populations. Moreover, the authors demonstrated that an exercise regimen was large enough to overcome an inefficient innervation. Finally, the same conditions were tested with human-derived MuSCs and MRCs, obtaining human muscle fibers in vivo.

As an alternative to stem cells isolation, a strategy to restore deficits could be represented by the employment of minced skeletal muscle from a donor site. This minced muscle could be straightforwardly implanted in a fresh VML wound or embedded in a matrix of collagen or a hyaluronic-acid-based hydrogel. Experiments conducted on rat models of VML reported an increment in new muscle formation in the compromised area and a partial restoration of force and functionality [3,55,56]. Another innovative approach regarding skeletal muscle tissue engineering for VML restoration is three-dimensional (3D) bioprinting. This technology allows highly organized and cellularized constructs with highly packed parallel fibers to be generated, reproducing the architectural organization of the native skeletal muscle (fundamental for its functionality) in a quick and reproducible manner. More specifically, the use of microfluidic techniques allows printing fibers with a remarkable resolution (printing fiber size) to be obtained. The work of Costantini and colleagues [1] revealed the potential of microfluidic-enhanced bioprinting in combination with a bioink composed of C2C12 (mouse myogenic progenitors) embedded in a PF–alginate hydrogel mix in driving proper differentiation of muscle fibers both in vitro and in vivo, with an impressive amelioration in terms of orientation and fiber organization. Moreover, very recently, the same authors [57] demonstrated the real potentiality of 3D printing technology in obtaining macroscopic myo-substitutes in vitro and in vivo, showing an astonishing functional and morphological recovery of TA VML mouse models (90% of TA muscle mass ablation) within just 3 weeks, revealing muscular tissue replenishment with new myofibers, blood vessels and nerves and promoting full TA muscle restoration in terms of strength performance. Moreover, the employed 3D printing technique turned out to be suitable for human myogenic cells. In another study, Choi and colleagues used ECMs obtained from skeletal muscle and porcine aorta to bioprint constructs of human skeletal muscle cells and human umbilical vessel endothelial cells (HUVECs), respectively. Specifically, they created printed constructs of both muscle cells alone and a combination of muscle and endothelial cells mixed together or a combination of the two populations spatially ordered by the use of a coaxial extruder. The latter allowed the printing of the shell of an endothelial-cells-laden hydrogel and the core of a muscle-laden hydrogel. The constructs were implanted in a rat model of VML, and the results obtained showed an increment in the formation of muscle fibers, isometric torque, and contractile force in artificial muscle. The results showed that the maximum muscle reformation was reached with the mixed construct coaxially printed [25]. A more recent work of Kim and colleagues investigated the role of neural cells in skeletal muscle regeneration. Bioprinted constructs of human muscle progenitor cells (hMPCs) with or without human neural stem cells (hNSCs) were implanted in a rat model of VML and analyzed by immune histological and functional analysis. The results highlighted the importance of the co-culture to ensure the best outcomes in terms of muscle weight, force, number of muscle fibers, and neuromuscular junctions in the regenerated muscle [58]. Nevertheless, the size still represents a hindrance to overcome.

### 5.2. Large Animal Models

Nowadays, the number of preclinical studies on large VML animal models is still limited. In fact, complications due to cell numbers, the amount of biomaterials, the size of artificial muscle correlated with problems on graft survival (early vascularization vs. tissue necrosis) and without a doubt the high costs make experimentation on this field a privilege that few research groups can afford. However, in recent years, some progress has been accomplished: Ward and collaborators have used an approach developed on rodents that consists in the implantation of autologous minced skeletal muscle on a VML site [59]. That is, a deficit of 20% was performed on the peroneus tertius muscle of female Yorkshire Cross swine, and within 30 min from excision of donor muscle, it was minced and transplanted into the injury site. The VML injuries caused a strength deficit of about 40%; and, after 12 weeks from implantation, the recovery in strength was about 30% in contrast with that of nonrepaired muscles. Histological analysis revealed the presence of neo-formed muscle fibers in implanted areas. Moreover, to better clarify the role of immune response to muscle regeneration, a similar approach was conducted in the presence of Tacrolimus (an immunomodulator), obtaining just a marginal force recovery (<20%) [60]. Taken together, these papers demonstrate that an approach based on autologous minced muscle grafts is able to restore muscle morphology and strength only partially.

Novakova and colleagues, instead, tested the regenerative capacity of their approach on an ovine model. Ovine muscle progenitors were cultured in monolayers and then delaminated to form a cylindrical structure with about 14 cm in length, called skeletal muscle unit (SMU), while ovine bone morrow stromal cells were cultured to create cylindrical hollow constructs, an engineered neural conduit (ENC). Then, VML injuries were performed on peroneus tertius of Polypay sheep, with or without the dissection of peroneal nerve to simulate nerve injury. SMU constructs were implanted in VML models, while SMU and ENC were implanted in VML and nerve damage models. After three months, both SMU- and SMU–ENC-implanted animals showed an increase in tetanic force, compared with the control. Furthermore, the immune histological analysis revealed the presence of muscle fibers and neuromuscular junctions at the VML site, indicating the amelioration of the regeneration process, despite the large amount of fibrotic tissue in all experimental conditions [61].

## 6. Discussion

Soft tissue injuries, especially skeletal muscle ones, are very common in daily life. Besides military personnel exposed to a wide series of combat-related trauma [6], civilian categories such as athletes, construction workers, or simply people behind the wheel are frequently victims of these kinds of incidents [27,28]. Unfortunately, VML occurs in a large percentage of muscle-related trauma, often leading to the development of chronic disability [29]. Actual therapies consist in wound debridement and surgical reconstruction by using free muscle flaps and physical training [41]. However, in the majority of the cases, these approaches are unsatisfactory, and the recovery of both aesthetically and functionality is completely inadequate [42]. For these reasons, there is a need for reconstructive therapies based on skeletal muscle tissue engineering. Whereas preclinical studies on animal models are very promising, especially those conducted on rodents, actual clinical treatments based on acellularized scaffolding are not enough to achieve a promising therapeutic approach [43,44,45,46]. Thus, the real challenge today is still the jump up from these cells-based therapeutic strategies to human size. At the present time, studies on large animal models are few, and the preliminary outcomes are not at all encouraging [59,60,61]. In addition, although the literature is exhaustive on the issues related to VML in terms of incidence [5,6,7], implication, current therapies, and emerging reconstructive strategies, some aspects need to be further investigated at a scientific level, such as consequences of volumetric loss on the socket of the prosthesis, the reduction of the contact surface on a prosthesis, and effects of re-innervation tagging on a reconstructed mass (Figure 1). We are confident that skeletal muscle tissue engineering is the right way to resolve the highly disabling pathology that negatively affects the quality of life of people suffering from VML-related pathologies. Furthermore, this reconstructive approach would be notably useful for replenishing prosthesis sockets and then enhance contacts and innervation surfaces for functional amelioration.

However, the translation of tissue engineering strategies to clinical practice is still a challenging task. In particular, there are three main limitations to overcome: (i) finding the optimal muscle progenitor source that show both myogenic potential and high proliferation rates to obtain a sufficient amount of cells; (ii) achieving a 3D tissue with an adequate density, dimensions and cell alignment to be comparable with a native muscle tissue architecture; (iii) promoting the in vivo integration and survival of an implanted tissue through rapid vascularization and innervation [62].

## Figures and Tables

**Figure 1 biomedicines-09-00564-f001:**
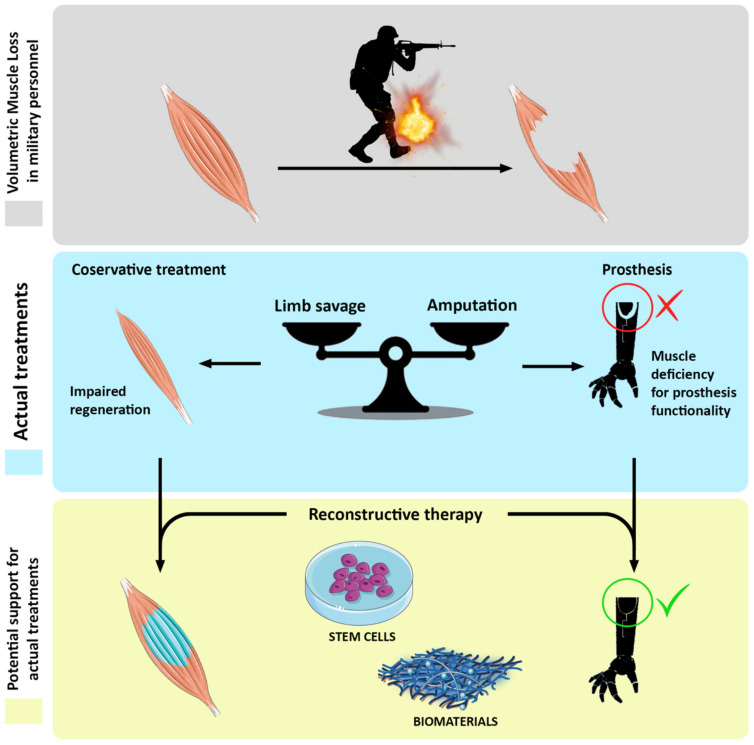
Schematic representation of cell-based reconstructive approach to volumetric muscle loss (VML) recovery.

## Data Availability

Not applicable.

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
