# Peer review of "The War after War: Volumetric Muscle Loss Incidence, Implication, Current Therapies and Emerging Reconstructive Strategies, a Comprehensive Review"

_biomedicines, 2021, doi:10.3390/biomedicines9050564_

Round 1
Reviewer 1 Report
The work from Testa et al., consist on a complete review of the events causing Volumetric Muscle Loss and possible therapeutic approach.
I have two one major comment.
1. The title should mention that this is a review.
2. Volumetric muscle loss can be also related to genetic myopathies that manifest with muscle atrophy and hypotrophy.
I suggest to add a a pargaraph describing these pathology and the contribution of celle therapies to treat these disorders.
Author Response
Comments and Suggestions for Authors
The work from Testa et al., consist on a complete review of the events causing Volumetric Muscle Loss and possible therapeutic approach.
I have two one major comment.
- The title should mention that this is a review.
The title has been modified according with the Reviewer.
- Volumetric muscle loss can be also related to genetic myopathies that manifest with muscle atrophy and hypotrophy.
I suggest to add a a pargaraph describing these pathology and the contribution of celle therapies to treat these disorders.
We thank the Reviewer for the suggestion, while respecting his/her opinion, the volumetric muscle loss is well known in literature to be an abrupt and bulk muscle tissue loss of traumatic nature (Corona et al., 2016, Kim et al., 2016, Basten et al., 2020, Bursac et al., 2016, Greising et al., 2019). In the work from Bursac and colleagues (Busac et al., 2016), genetic myopathies, VML and diseases that result in muscle atrophy and wasting are divided into separate categories of muscle disorders. Greising and colleagues also specify that VML is a form of muscle trauma that differs in etiology from progressive conditions of muscle loss associated with aging or disease (Greising et al., 2019). The intent of this review is to deepen the aspects concerning VML, consequently adding a separate paragraph about other types of muscle disorders, despite interesting, fall outside the aim of this review. We do realize that we misled the reader including “chronic diseases” as a possible cause of VML in the abstract, so we modified the section removing the term.
Reviewer 2 Report
Thank you for this well written, informative and structured article which reaches out to address an important issue (VML). You well described the circumstance leading to these injuries as well as the current treatment strategies.
This article would be of great interest to the readership however would substantial modificant would be necessary. This issue (VML) should be addressed through a systematic review (also see PRISMA guidelines) to ensure and fulfill all necessary requirements and furthermore reaches out to all data base (EMBASE, Medline, etc..).
Author Response
Comments and Suggestions for Authors
Thank you for this well written, informative and structured article which reaches out to address an important issue (VML). You well described the circumstance leading to these injuries as well as the current treatment strategies.
This article would be of great interest to the readership however would substantial modificant would be necessary. This issue (VML) should be addressed through a systematic review (also see PRISMA guidelines) to ensure and fulfill all necessary requirements and furthermore reaches out to all data base (EMBASE, Medline, etc..).
We thank the reviewer for the suggestion, as we were not aware of database regarding specific protocols for systematic reviews. This review was not conceived from the beginning with a systematic approach, but we will certainly consider the suggested guidelines for future works.
“A guideline to help authors prepare protocols for planned systematic reviews and meta-analyses that provides them with a minimum set of items to be included in the protocol. A protocol is intended to provide the rationale for the review and pre-planned methodological and analytic approach, prior to embarking on a review. Investigators should prepare a review protocol in advance of registering it in PROSPERO so that details requiring further consideration may be thought through in advance, avoiding the need for multiple amendments to registration information.” from Moher et al., 2015.
ADMINISTRATIVE INFORMATION |
||
Title |
||
Identification |
1a |
Identify the report as a protocol of a systematic review |
Update |
1b |
If the protocol is for an update of a previous systematic review, identify as such |
Registration |
2 |
If registered, provide the name of the registry (e.g., PROSPERO) and registration number |
Authors |
||
Contact |
3a |
Provide name, institutional affiliation, and e-mail address of all protocol authors; provide physical mailing address of corresponding author |
Contributions |
3b |
Describe contributions of protocol authors and identify the guarantor of the review |
Amendments |
4 |
If the protocol represents an amendment of a previously completed or published protocol, identify as such and list changes; otherwise, state plan for documenting important protocol amendments |
Support |
||
Sources |
5a |
Indicate sources of financial or other support for the review |
Sponsor |
5b |
Provide name for the review funder and/or sponsor |
Role of sponsor/funder |
5c |
Describe roles of funder(s), sponsor(s), and/or institution(s), if any, in developing the protocol |
INTRODUCTION |
||
Rationale |
6 |
Describe the rationale for the review in the context of what is already known |
Objectives |
7 |
Provide an explicit statement of the question(s) the review will address with reference to participants, interventions, comparators, and outcomes (PICO) |
METHODS |
||
Eligibility criteria |
8 |
Specify the study characteristics (e.g., PICO, study design, setting, time frame) and report characteristics (e.g., years considered, language, publication status) to be used as criteria for eligibility for the review |
Information sources |
9 |
Describe all intended information sources (e.g., electronic databases, contact with study authors, trial registers, or other grey literature sources) with planned dates of coverage |
Search strategy |
10 |
Present draft of search strategy to be used for at least one electronic database, including planned limits, such that it could be repeated |
Study records |
||
Data management |
11a |
Describe the mechanism(s) that will be used to manage records and data throughout the review |
Selection process |
11b |
State the process that will be used for selecting studies (e.g., two independent reviewers) through each phase of the review (i.e., screening, eligibility, and inclusion in meta-analysis) |
Data collection process |
11c |
Describe planned method of extracting data from reports (e.g., piloting forms, done independently, in duplicate), any processes for obtaining and confirming data from investigators |
Data items |
12 |
List and define all variables for which data will be sought (e.g., PICO items, funding sources), any pre-planned data assumptions and simplifications |
Outcomes and prioritization |
13 |
List and define all outcomes for which data will be sought, including prioritization of main and additional outcomes, with rationale |
Risk of bias in individual studies |
14 |
Describe anticipated methods for assessing risk of bias of individual studies, including whether this will be done at the outcome or study level, or both; state how this information will be used in data synthesis |
Data |
||
Synthesis |
15a |
Describe criteria under which study data will be quantitatively synthesized |
15b |
If data are appropriate for quantitative synthesis, describe planned summary measures, methods of handling data, and methods of combining data from studies, including any planned exploration of consistency (e.g., I2, Kendall’s tau) |
|
15c |
Describe any proposed additional analyses (e.g., sensitivity or subgroup analyses, meta-regression) |
|
15d |
If quantitative synthesis is not appropriate, describe the type of summary planned |
|
Meta-bias(es) |
16 |
Specify any planned assessment of meta-bias(es) (e.g., publication bias across studies, selective reporting within studies) |
Confidence in cumulative evidence |
17 |
Describe how the strength of the body of evidence will be assessed (e.g., GRADE) |
As stated above the pre-planned methodological approach above reported must be conceived prior of review drafting, nevertheless most of the listed issues were taken into account for the review editing.
We thank the reviewer for Embase hint, we did not know this database, but at the end of the day the difference with Pubmed (we normally use for literate searching) are the conference abstracts that often are overlapping with future papers.
Reviewer 3 Report
This is an interesting review from the Gargioli lab on volumetric muscle loss.
Minor comments:
1-The manuscript would gain from a thorough discussion about limitations of on tissue engineering-based approaches in human.
2-Could the authors add some discussion about the molecular mechanisms leading to HO? Are there evidence that tissue engineering-based therapies prevent such complication?
Author Response
Comments and Suggestions for Authors
This is an interesting review from the Gargioli lab on volumetric muscle loss.
Minor comments:
1-The manuscript would gain from a thorough discussion about limitations of on tissue engineering-based approaches in human.
We thank the Reviewer his/her criticism, accordingly a section about tissue engineering limitations has been added to the discussion.
2-Could the authors add some discussion about the molecular mechanisms leading to HO? Are there evidence that tissue engineering-based therapies prevent such complication?
Thanks to the Reviewer’s suggestion, some molecular aspects of HO have been added to the text. Although it would be very interesting to deepen this topic, to date we have not been able to collect sufficient bibliographic sources that link tissue engineering approaches and heterotopic ossification.
Reviewer 4 Report
The reviewer would like to thank the authors for writing down this review. The topic is really interesting and important. However, during the reviewing process, some points have been raised which require modification.
CORRESPONDENCE
Better mention only one author for correspondence.
ABSTRACT
Upon VML occurring, the affected area is severely compromised, heavily conditioning the affected person’s quality of life: affecting the patient's quality of life.
Keywords: VML; war muscle injuries; permanent disability; prosthesis; reconstructive therapies: Please add muscle loss, war, muscle injury
- The authors are going back and forth to the causes of muscle injury. This could be a heading reporting the different causes of muscle injury as it's written in several paragraphs which could be gathered in one paragraph emphasizing the injuries related to the war.
War related skeletal muscle injuries
area with associated leak of stem cells: There is no leak but loss of stem cells.
Social implication of muscle-related disabilities
Extremity injuries are one of the most common wounds among military personnel, accounting for a high percentage of all injuries sustained in military combat operations: This is just a repetition of the previous paragraph. Please correct!
Talbot and colleagues noticed that, whereas, physical health improved as might be expected upon rehabilitation, mental health component did not[19]: Please check reference number 19 as it's col et al not Talbot!
When reintegration is not possible, the soldier is medically retired and the conditions .......disability of military personnel[6]: This sentence is again a repetition of sentences of previous paragraphs.
Current treatments of VML
Surgical treatments
VML injuries occur following a large variety of trauma ranging from high....battlefields and in many cases is: Again the authors are repeating the sentences.
Nevertheless, the size still represents a to overcome hindrance: Please check the grammar.
Corona and collaborators have used an approach developed on rodents ..... on VML site[58]: Please correct this reference as it's not Corona but Ward, C.L.
It's better if you mention an authors et al to put the reference directly next to it or if you want to put the reference at the end of the text or paragraph, just mention a study was conducted reporting..... then reference at the end.
DISCUSSION
Please add the citations and references to the discussion as it's missing all the references.

Author Response
Comments and Suggestions for Authors
The reviewer would like to thank the authors for writing down this review. The topic is really interesting and important. However, during the reviewing process, some points have been raised which require modification.
CORRESPONDENCE
Better mention only one author for correspondence.
Respecting Reviewer opinion, because of different area if competence we believe that two corresponding authors are necessary for this review.
ABSTRACT
Upon VML occurring, the affected area is severely compromised, heavily conditioning the affected person’s quality of life: affecting the patient's quality of life.
Modified
Keywords: VML; war muscle injuries; permanent disability; prosthesis; reconstructive therapies: Please add muscle loss, war, muscle injury
Added
- The authors are going back and forth to the causes of muscle injury. This could be a heading reporting the different causes of muscle injury as it's written in several paragraphs which could be gathered in one paragraph emphasizing the injuries related to the war.
War related skeletal muscle injuries
Modified
area with associated leak of stem cells: There is no leak but loss of stem cells.
Corrected as suggested.
Social implication of muscle-related disabilities
Modified
Extremity injuries are one of the most common wounds among military personnel, accounting for a high percentage of all injuries sustained in military combat operations: This is just a repetition of the previous paragraph.
Corrected
Talbot and colleagues noticed that, whereas, physical health improved as might be expected upon rehabilitation, mental health component did not[19]: Please check reference number 19 as it's col et al not Talbot!
We thank the Reviewer for the hint, we corrected the mistake.
When reintegration is not possible, the soldier is medically retired and the conditions .......disability of military personnel[6]: This sentence is again a repetition of sentences of previous paragraphs.
Current treatments of VML
Surgical treatments
VML injuries occur following a large variety of trauma ranging from high....battlefields and in many cases is: Again the authors are repeating the sentences.
Nevertheless, the size still represents a to overcome hindrance: Please check the grammar.
We checked the sentence.
Corona and collaborators have used an approach developed on rodents ..... on VML site[58]: Please correct this reference as it's not Corona but Ward, C.L.
The reference has been corrected.
It's better if you mention an authors et al to put the reference directly next to it or if you want to put the reference at the end of the text or paragraph, just mention a study was conducted reporting..... then reference at the end.
DISCUSSION
Please add the citations and references to the discussion as it's missing all the references.
We insert references in the discussion, agreeing with the reviewer suggestion.
We thank the reviewer for the observations. Accordingly, we corrected the indicated blunders and revised the manuscript in a critical way. We do realize that some paragraphs begin with sentences describing aspects previously discussed, but in a way that in our opinion should help the reader to introduce a deeper analysis of the topic while keeping some important concepts.
Round 2
Reviewer 2 Report
Thank you for your modifications and to consider more data base information for future trials. Well prepared article with sound information and comprehensive information for a wide variety of medical specialists.